# SBAS-InSAR Based Deformation Monitoring of Tailings Dam: The Case Study of the Dexing Copper Mine No.4 Tailings Dam

**DOI:** 10.3390/s23249707

**Published:** 2023-12-08

**Authors:** Weiguo Xie, Jianhua Wu, Hua Gao, Jiehong Chen, Yufeng He

**Affiliations:** 1Key Laboratory of Poyang Lake Wetland and Watershed Research, Ministry of Education, School of Geography and Environment, Jiangxi Normal University, Nanchang 330022, China; xiewg@jxnu.edu.cn (W.X.); chenjiehong@jxnu.edu.cn (J.C.); yuf.he@jxnu.edu.cn (Y.H.); 2Key Laboratory of Natural Disaster Monitoring, Early Warning and Assessment of Jiangxi Province, Jiangxi Normal University, Nanchang 330022, China

**Keywords:** SBAS-InSAR, tailing dam, Sentinel-1, deformation monitoring, copper mine

## Abstract

The No.4 tailings pond of the Dexing Copper Mine is the second largest in Asia. The tailing pond is a dangerous source of man-made debris flow with high potential energy. In view of the lack of effective and low-cost global safety monitoring means in this region, in this paper, the time-series InSAR technology is innovatively introduced to monitor the deformation of tailings dam and significant key findings are obtained. First, the surface deformation information of the tailings pond and its surrounding areas was extracted by using SBAS-InSAR technology and Sentinel-1A data. Second, the cause of deformation is explored by analyzing the deformation rate, deformation accumulation, and three typical deformation rate profiles of the representative observation points on the dam body. Finally, the power function model is used to predict the typical deformation observation points. The results of this paper indicated that: (1) the surface deformation of the tailings dam can be categorized into two directions: the upper portion of the dam moving away from the satellite along the Line of Sight (LOS) at a rate of −40 mm/yr, whereas the bottom portion approaching the satellite along the LOS at a rate of 8 mm/yr; (2) the deformation of the dam body is mainly affected by the inventory deposits and the construction materials of the dam body; (3) according to the current trend, deformation of two typical observation points in the LOS direction will reach the cumulative deformation of 80 mm and −360 mm respectively. The research results can provide data support for safety management of No.4 tailings dam in the Dexing Copper Mine, and provide a method reference for monitoring other similar tailings dams.

## 1. Introduction

Tailings ponds are the places used to store tailings or other industrial wastes released from metallic or non-metallic mines, which are generally formed by constructing dams at the valley entrance. They are often distributed near mines, and their reservoir area can reach several square kilometers [1]. Tailings dams are dams built for the accumulation of various tailings, and their height can reach hundreds of meters [2]. In the events of dam failures, a tailings dam is a kind of man-made debris flow hazard and pollution source with high potential energy. Once the dam breaks, it will seriously affect the safety of life, property, and the ecological environment of downstream residents [1]. The causes of a tailings pond dam break usually fall into the following categories: static failure, seepage, internal erosion, structural and foundation conditions, overload, structural flaws, seismic instability, mine subsidence and external erosion, etc. [3]. In recent years, there have been many major accidents caused by the damage of tailings dams worldwide [4]. The major tailings dam break events in the world in recent years are shown in Table 1. On 25 January 2019, due to torrential rain the tailings impoundment at Córrego do Feijo mine of Vale do Rio Doce in Brazil was breached, resulting in more than 200 people dead or missing [5]. On 26 November 2021, the tailings dam in San Antonio de Maria mine, Peru failed after heavy rain, the tailings pond sedimentation tank leaked, and the leakage of arsenic and cyanide caused serious environmental pollution [6]. Shanxi Daoer Aluminum (Lvliang City, Shanxi Province) experienced a dam failure on 27 March 2022 due to the phenomenon of “melting and sinking” of the dam soil, caused by spring warming and rainfall, and a decrease in the dam’s load-bearing capacity. The accident caused the direct burial of a portion of a business’s facility, more than 200 m of seasonal ditches, and rural roadways directly beneath the tailings pond [7]. Globally, an average of two to five large tailings ponds fail annually, according to statistics [8], and the probability of tailings pond failure is approximately ten times that of reservoir dam failure [9]. Therefore, monitoring the stability of tailings dams is essential for the safety of tailings ponds and the long-term development of mines [10].

Dam deformation is an important indicator reflecting the stability of tailings dams. Regular deformation monitoring and analysis of the deformation characteristics of open-pit mines and sediment dams can identify potential problems in time and effectively avoid risks [25]. Currently, monitoring methods for tailings dam deformation are mainly divided into two categories: geotechnical engineering instruments and non-geotechnical instruments methods. The former includes hydrometers, extensometers, shape accelerometers, inclinometers, and water level indicators, etc. [26,27]. The latter includes total station/reflecting prism, video monitoring, etc. [28,29]. The above methods are usually used for single point monitoring of sediment dams. Although the data provided by these instruments are accurate and reliable, there are some limitations to using them to observe data. For instance, they are severely affected by weather, limited instrument field of view, or operating range [30,31,32]. The most critical issue is that the aforementioned methods can only monitor specific points or parts on the dam body, and cannot acquire information on the dam’s overall deformation [33].

Interferometric Synthetic Aperture Radar (InSAR) is an all-weather, round-the-clock, wide-coverage, high-resolution, and high-precision technique for monitoring the stability of artificial structures. InSAR does not require ground control points and can be observed without fieldwork. Consequently, it can significantly reduce observation costs while covering a large area with intensive measurements and providing high-precision surface deformation data [34]. Since 1989, when the D-InSAR technique was first used to detect surface movement [35], InSAR has been extensively utilized to monitor various types of surface settlements [36,37,38], including various types of dams and the deformation of their surrounding areas [39,40,41,42]. Multi-temporal Interferometric Synthetic Aperture Radar (MT-InSAR) is a technique to obtain high-precision temporal deformation information of the surface by combining multiple landscape survey images to suppress the influence of atmospheric errors and noise. The PS-InSAR (Persistent Scatterer InSAR) method [43] and the SBAS-InSAR (Small Baseline Subset InSAR) method [44] are the most commonly used MT-InSAR methods. Surface deformation of open-pit slopes, tailings basins, and tailings dams can be effectively monitored [9,27,45,46]. PS-InSAR is more sensitive to stable scatterers; this method uses the phase and amplitude information of the same coverage SAR images in the same area of multiple views to select point targets that are not affected by temporal and spatial de-correlation for deformation extraction. On the basis of statistical analysis, PS-InSAR can mitigate the effects of errors such as atmospheric errors for point targets, thereby estimating the sluggish successive movement rates and temporal displacements at each PS point [47,48,49,50,51]. SBAS-InSAR mitigates the effects of spatial-temporal de-correlation by selecting spatial-temporal baseline pairs of multiple primary image components with short baseline lengths. The technique is applicable to a wide variety of surfaces and excels in extracting deformation from non-urban areas in densely vegetated regions [30,52,53,54].

The SBAS-InSAR technique for monitoring surface deformation is relatively mature. This technology can monitor large-scale deformation with millimeter-level accuracy, and has been widely used in various geodetic fields such as land subsidence, landslide, and seismic activity [55,56,57]. Some scholars have also applied it to the deformation monitoring research of tailings dams, but the majority concentrate on the deformation analysis of the dam body after dam failure, and very few investigate the deformation of tailings dams in operation. Current research on the Dexing Copper Mine concentrates primarily on copper mining, the impact of mining on the ecological environment, mine slope management, mine geological information, and changes of mining surface vegetation cover [58,59,60,61,62]. A few scholars have studied the deformation of the Dexing Copper Mine area, including the open pit, plant, and supporting areas [63], but there is a lack of research on the stability of the tailings dam.

In this study, the SBAS-InSAR technique was chosen for the deformation extraction of the No.4 tailings dam of the Dexing Copper Mine. The high-precision and long-time-series information of the dam deformation is obtained based on 118 Sentinel-1A images from 2018 to 2021. Based on the deformation results, the deformation rate map of the dam and the deformation accumulation map are plotted, which are used to evaluate the stability of the dam. The power function fitting model is used to predict the future deformation trend of the tailings dam, which can provide data to the pertinent departments.

## 2. Study Area

Dexing City is located in Jiangxi Province, China, on the southeastern edge of the Degu Land in southern Jiangnan, and on the transitional zone between the Zhe-Gan depression and the northwestern side of the Northeast Gan Deep Rift. The main outcrops in the region are the shallow metamorphic facies of the Shuangqiaoshan Group of the Aurignacian System [59]. The subtropical monsoon climate zone of Dexing is mild and humid, with an average annual temperature range of 16–19 °C and an average annual precipitation of 1800 mm [64]. The Dexing Copper Mine is the largest open-pit copper mine in Asia, with a proven copper reserve of 1.63 billion tons and an annual mine production of 160,000 tons of copper, representing approximately one-fifth of the nation [58]. At the border of the Dexing Copper Mine open pit, the No.4 tailings pond was constructed, with the No.4 tailings pond having the largest volume, the largest surface area, and the largest dam volume (Figure 1) [62].

The No.4 tailings pond is located in the Xiyuada ditch, which is surrounded by mountains on three sides. The terrain is elevated in the south and low in the north, and the main ditch in the center is approximately 5 km long and runs north–south. The tailings pond has a combined capacity of 835 million cubic meters and a catchment area of 14.3 km^2^. Figure 1c depicts the main portion of the No.4 tailings pond, in which the dry beach is in the blue line, the blue-green water surface on the south side of the dry beach is the copper bearing wastewater of the tailings pond, the top of the No.4 tailings dam is in the green line, the outer slope range is in the orange line, and Shiduntou village is in the northernmost pink line. To analyze the deformation characteristics of the tailings dam in detail, the outer slope is divided into a bottom part and an upper part based on its geographic location, which is bounded by the east–west center line of the outer slope. The scope of the bottom part extends from the center line of the outer slope to the south end of the seepage cut-off pool at the bottom of the slope, while the scope of the upper part extends from the center line of the outer slope to the trough.

The construction of the tailings dam at Tailings Storage Facility No.4 was divided into two phases: an initial dam and an accumulation dam (Figure 2). The dam body, constructed between 1988 and 1990, is composed of medium-weathered fresh phyllite. It is important to note that the tailings dam, the first of its kind in China to be built using the centerline method, was erected vertically along the initial dam. Designed to be built in eleven phases from 1991 to 2008, it served as a significant achievement [66]. The upper part of the site’s geological structure consists of water and artificial fill. The lower part is composed of the Pre-Cambrian Double Bridge Mountain Group bedrock. Based on its lithology and engineering characteristics, the division from top to bottom is as follows: Water, Unaltered fill soil, Colluvial fill soil, Strongly weathered tuffaceous sandstone, and Moderately weathered tuffaceous sandstone. The tailing ore is pale yellow-pink granules. The copper content in the tailings ore is 0.139%; it also contains other valuable gold groups such as Fe, Pb, Zn, etc. The size of the tailings is fine, and the tailings with a diameter of −38 μm account for 65%. The pH of the supernatant of the tailing pulp is in the range of 11.5 to 12.5, the pH value is high, and the content of organic matter is less. After settling, the contents of Fe, Ca, Al, and Mg in the tailings are higher, while the contents of Pb and Cd are less than 5 μg/g. The pH of acidic wastewater of mine mountain is in the range of 2.0 to 2.3, the pH value is low, and the waste water contains a lot of heavy metal elements [67].

The dam information is shown in Table 2. The initial dam is situated in the middle of the current tailings dam’s axis, and its axis is at an angle of 15.70° to the axis of the final dam of the later stockpile. The initial dam crest elevation is 110 m, the dam height is 38 m, crest width is 10 m, and the external slope ratio is 1:3. The initial dam slope is rendered impermeable by a clay inclined wall, the base of which is formed into a tooth groove. All tailings are first classified in a two-stage classification cyclone, with the coarse sand used for piling and the slurry discharged into the tailings pond [68]. The summit of the dam has a current elevation of 280 m, a length of approximately 1350 m, a width of 40 m, and an elevation of 272.79 m at its highest point. The length of the dry shore in the tailings storage reservoir is 2251.78 m, while the reservoir’s normal storage level is 266.6 m. The north side of the dam body is equipped with a 9 m tall slurry block dam. The elevation of dam’s summit is 80 m, and the elevation of dam’s base is 68 m, and the width of the dam’s crest is 2 m (Figure 2). The No.4 tailings dam has a surface drainage system, a seepage system, and two flood discharge systems. Three seepage interceptors with depths between 0.9 m and 1.4 m are located at the base of the outer slope of the No.4 tailings dam. About 100 m to the north of the seepage interceptors is the Shiduntou group of Ducun Village, Dexing City, which consists of 460 households and 1700 individuals [69].

## 3. Materials and Methods

### 3.1. Data

Sentinel-1 is a C-band SAR satellite system developed by the European Space Agency (ESA). The system consists of two satellites, Sentinel-1A and Sentinel-1B, which were launched in 2014 and 2016, respectively (Sentinel-1B was terminated due to failure on 23 December 2021). The system revisit period is 12 days for single stars and 6 days for binary stars. The Sentinel-1 TOPS model has a spatial resolution of 5 m × 20 m and a coverage area of 250 km, which is very suitable for monitoring the deformation of a large-scale surface.

Due to the limited presence of the Sentinel-1 constellation with only one satellite ascending orbit passing through the study area (The location of the No.4 tailings pond), we acquired nearly 4 years of data from the Sentinel-1 satellite (Path142-Frame91) spanning from 3 January 2018 to 13 December 2021, including a total of 118 images with a revisit interval of 12 days. The main parameters of these images are shown in Table 3.

Additionally, to remove the topographic phase and acquire accurate deformation information, we employed NASA’s SRTM (Shuttle Radar Topography Mission) DEM (Digital Elevation Model) [70]. Furthermore, to explore the relationship between tailings dam deformation and rainfall, we utilized monthly mean rainfall data from the Tibetan Plateau Data Center. The data include rainfall data from all meteorological stations in China, which combines with bilinear interpolation of slope, elevation, and other topographic information.

### 3.2. Methodology

Because the study area features intricate topography, primarily comprising mountains, hills, basins, and water bodies, with few permanent scatterers, we employed the SBAS-InSAR technology to monitor the deformation of the Dexing Copper No.4 tailings dam. SBAS-InSAR can cover the surface of the study area more thoroughly by uniformly extracting the distributed scatterers, allowing for a more thorough detection of surface changes [71]. This technique is a time-series InSAR technique based on D-InSAR. It generates multiple interferograms by setting the spatial and temporal ranges of the baseline, performs multiview processing on the interferograms to enhance the phase quality of the interferograms, optimizes the phase unwrapping process, and mitigates the effects of spatial and temporal de-correlation as well as atmospheric disturbances by utilizing the spatial and temporal information of the multipair interferograms. The phase component of each pixel in the interferogram includes not only the settlement phase, but also terrain, atmosphere, orbit error, and noise. The phase component can be expressed as follows:(1)$$\varphi_{\Delta t\left(x,r\right)=}\varphi_{dr\left(x,r\right)}+\varphi_{h\left(x,r\right)}+\varphi_{atm\left(x,r\right)}+\varphi_{\beta \left(x,r\right)}+\varphi_{n\left(x,r\right)}$$
where *x* and *r* are the azimuthal and distance coordinates; $\varphi_{dr}$ is the phase shift due to pixel displacement in the Line of Sight (LOS) direction; $\varphi_{h}$ is the terrain phase error, which is removed by introducing an external DEM; $\varphi_{atm}$ is the atmospheric phase delay, which is attenuated by spatio-temporal filtering; $\varphi_{\beta }$ is the residual phase due to orbital error, which is removed by introducing POD (Precise Orbit Ephemerides); and $\varphi_{n}$ is the phase noise, which is eliminated by spatial filtering with a specified threshold.

We then utilize Singular Value Decomposition (SVD) to solve the least-squares solution [54] and derive the azimuthal phase averaging rate. Multiple SAR images of the same region acquired at different time intervals and with minor geometric differences are analyzed to produce differential interferograms, which enable the detection of temporal variations of ground displacement in the LOS direction. The data processing flowchart of SBAS-InSAR is shown in Figure 3.

As shown in Figure 3, data processing mainly includes five steps:

Step 1: firstly, in order to reduce image noise and reduce data volume, we need to perform 4:1 (range: azimuth) multi-look processing on 118 images; then we obtained 118 new images with a spatial resolution of about 20 m. Secondly, due to the rugged terrain and dense vegetation distribution in the study area, the decoherence effect is obvious; we established a temporal baseline threshold of 48 days and a spatial baseline threshold of 200 m. Finally, we generated a total of 339 pairs of interferometric baselines; the baseline network is shown in Figure 4.

Step 2: during the process, to improve the orbital accuracy, we employed the POD. To remove the topographic phase, the SRTM DEM with a spatial resolution of 30 m is used. And Goldstein filtering is applied to interferograms for spatial filtering with a window of 64 and a filter factor of 0.3 [72].

Step 3: in the process, we employed the Minimum Cost Flow (MCF) algorithm to unwrap the spatial phase of the interferograms [73], then we obtained the unwrapping interferograms.

Step 4: the GACOS data (Generic Atmosphere Correction Online Service for InSAR [74]) and spatio-temporal filtering are used to attenuate the effect of the atmospheric phase. The linear least-squares model [75] is used to solve the temporal and spatial filtering results; the Singular Value Decomposition (SVD) method is used to extract the temporal surface subsidence from the unwrapped phase with all kinds of error removed.

Step 5: the surface subsidence data during the observation period are calculated, and the LOS deformation field in the study area is obtained and projected to the geographical coordinate system.

## 4. Result

The LOS surface deformation of tailings dam No.4 in the Dexing Copper Mine is shown in Figure 5. According to the strategy in 3.2, we acquired a total of 32,903 observation points. The study area is completely covered, the point density is high, and the distribution is uniform. In this study, the direction away from the satellite is called sinking (cool color), and the direction closer to the satellite is called lifting (warm color).

According to construction time and dam material, the outer slope of the dam body can be divided into an upper part and a bottom part, with the bottom part’s building time having started sooner than the upper part’s. As depicted in Figure 5, the observation points extracted using the SBAS-InSAR method cover most of the non-water body area. In Figure 5, warm colors indicate surface changes toward the satellite, while cool colors indicate surface changes moving away from the satellite. Based on the deformation analysis from January 2018 to December 2021, significant surface deformation occurred in the vicinity of the dam, which was primarily characterized by moving away from the satellite along the LOS (M-LOS) in the south-central portion, and approaching to the satellite along the LOS (A-LOS) in the northern and eastern portions. The deformation rates in the area range between −46 mm/yr and +13 mm/yr. The lowest value of −46 mm/yr is located near the dam’s crest (blue triangle in Figure 5), while the highest value (13 mm/yr) is located on a conspicuous ridge on the dam’s eastern slope (red triangle in Figure 5). At the junction of the upper and bottom portions of the dam body, relative stability exists, with deformation rates ranging from −1 to 1 mm/yr.

As shown in Figure 6a, the deformation rate profile of AA’ indicates that the lowest values of deformation rate are observed at the center of the top of the dam and 600 m north of the center. The deformation rate tends to be zero at the top and bottom of the dam. After the boundary line, the deformation rate at the bottom of the dam ranges from 5 to 10 mm/yr. The A-LOS area stretches northward towards the northern section of the seepage interceptor pool and southeast towards the natural hillside on the eastern side of the dam. According to the deformation rate profile at BB’ (Figure 6b), the dam body exhibits the lowest deformation rate of −43 mm/yr in its upper center. The hillslopes on the west side of the BB’ line experience more stable deformation, with rates ranging from −1 mm/yr to 2 mm/yr, whereas the east side of the hillslopes demonstrates deformation rates between 5 and 13 mm/yr, resulting in a A-LOS area on the eastern side of the dam. With a deformation rate ranging from 5 to 13 mm/yr, the east slope forms an A-LOS zone that extends from the dam crest towards the upstream region of the dam body. This A-LOS zone encircles the downstream tailings dam, stretching from east to north and spanning around 1300 m in length and 240 m in width. The CC’ line (Figure 6c) traverses the bottom of the dam body in a west-to-east direction, and the deformation rate within this pathway area exhibits considerable variation, initially increasing and then decreasing from west to east. The highest deformation rate, reaching 13 mm/yr, is found in the ridge that extends from the eastern base of the dam, coinciding with the aforementioned A-LOS zone.

On the basis of the deformation monitoring data from January 2018 to December 2021, we plotted the time-series deformation cumulative maps (Figure 7 and Figure 8) for representative observation points (P1–14 in Figure 5). These points, along with the positive extreme, negative extreme and the stabilization points mentioned earlier (triangular points in Figure 5), form a network covering the dam body and surrounding areas, covering all types of deformation.

The findings show that P1, P9, and P14 are situated in Shiduntou village on the north side of the seepage interceptor, the hillside on the west side of the dam body, and the stabilization zone in the middle of the dam body, respectively. P2 to P6 are distributed along the bottom of the dam body at roughly equal intervals, with the deformation rate ranging from 5 mm/yr to 12 mm/yr, and the cumulative amount of deformation ranging from 25 mm to 12 mm. This is the stabilization type, with a deformation rate ranging from −3 mm/yr to 1 mm/yr, and a cumulative amount of deformation ranging from −10 mm to 10 mm. The cumulative amount of A-LOS ranges from 25 mm to 50 mm, and they are generally located in the A-LOS zone to the north of the stabilized zone. P7, P8, P10, and P13 are located in the upper portion of the dam body, along with the top of the dam and the dry beach. The amount of M-LOS is larger and unevenly distributed, with deformation rates ranging from −17 mm/yr to 42 mm/yr and the cumulative amount of M-LOS ranging from −103 mm to 180 mm/yr. The deformation characteristics and trends of these locations are consistent with the overall deformation characteristics of the dam body in each of the regions described previously.

## 5. Discussion

### 5.1. Precision Checking and Impact of Errors

To estimate the precision of the deformation monitoring results, we calculated the overall fitting accuracy of the LOS toward the deformation rate in the study area [76] based on the fitted standard deviation of the linear rate.
(2)$$\sigma_{i}=\sqrt{\frac{\sum_{k=1}^{n}{\left({def}_{fit,k}-{def}_{inf,k}\right)}^{2}}{n}}$$
where $\sigma_{i}$ represents the standard deviation of the rate at point *i* and *n* represents the number of interferograms used to calculate the deformation rate. ${def}_{fit,k}$ represents the deformation of point *i* obtained from the $k_{th}$ interferogram, whereas ${def}_{inf,k}$ represents the deformation value of point *i* obtained by fitting with the deformation rate. According to the above equation, the standard deviation of each observation point is calculated, and the result is depicted in Figure 9a.

According to the statistical analysis of each point’s standard deviation (Figure 9b), we were able to estimate the deformation rate with a 95% confidence interval to be between 0.237 to 0.785 mm/yr. The fitting accuracies in areas such as the edge of the dam body and the bottom of the dam body are approximately 0.2~0.5 mm/yr; the fitting accuracies in the upper portion of the dam body are slightly lower than those in other areas, with the maximum value of 0.903 mm/yr located on the middle west side of the middle of the dam body and the middle north side of the dam top. Overall, the fitting accuracy of the mountain’s deformation rate is higher than that of the mountain’s edge and the bottom of the dam body, and we hypothesize that this is due to the increased construction intensity of the upper portion of the dam body.

We utilized a total of 118 Sentinel-1A images over four years and set a shortened spatio-temporal baseline, and the SBAS-InSAR technique produced more accurate deformation results, with an overall internal fitting accuracy of less than 1 mm/yr (Figure 9). However, errors cannot be completely avoided. First, for mountainous areas with dense vegetation, the coherence is poor [77]. During the observation period of this study, there was no large-scale construction of the tailings pond. However, due to the influence of rainfall and vegetation growth, some slight changes occurred on the surface of the outer slope, resulting in a certain incoherence. We were unable to completely eliminate the effects of spatio-temporal decoherence and atmospheric errors [78], despite establishing a 48-day temporal baseline and a 200-m spatial baseline and using GACOS atmospheric delay data. Errors caused by these effects eventually show up in the deformation monitoring results (Figure 5).

There is no public GNSS or leveling observation data in this region for comparison, so we cannot obtain accurate external coincidence accuracy of the InSAR deformation field. Although the accuracy evaluation method in this paper can not directly obtain the real error of the deformation rate field, our calculation results can reflect the time-series stability of the surface deformation of the tailings dam and the accuracy of the deformation rate to a certain extent. The deformation error caused by natural surface variations can be disregarded.

### 5.2. Cause Analysis of the Deformation in Dexing No.4 Tailings Dam

Figure 10 depicts the cumulative quantity of deformation from January 2018 to December 2021 for the three profile lines AA’, BB’, and CC’. From the AA’ profile line in Figure 10a, it can be seen that the cumulative quantity of M-LOS in the upper portion of the dam body is greater, reaching −170 mm, while the bottom of the dam body exhibits A-LOS, reaching 50 mm. There is a clear dividing line between the upper part of the dam body and the bottom, that is, the road between the upper part and the bottom part. The M-LOS of the upper portion of the dam body exhibits a uniform increase with time, and the deformation rate is only slightly slower in the first half of 2020 than at other times. However, the A-LOS of the lower portion of the dam body exhibits a fast and then slow increase with time, and the A-LOS rate is slightly faster before October 2019 than after this moment. From the BB’ profile line in Figure 10b, it can be seen that the cumulative amount of M-LOS in the upper part of the dam body is larger, reaching −150 mm, and that the rate of M-LOS is first fast, then slow, and then accelerated, with the same overall deformation trend, and that the center of deformation is located near the dam body’s midline, which is consistent with the trend of the distribution of the deformation rate in the previous profile line. From the CC’ profile in Figure 10c, the maximum cumulative amount of deformation at the bottom of the dam body is 58 mm, and the rate of lifting is first fast and then slow; the amount of lifting in the area on the north side of the dam body facing the seepage interceptor is less than that in the surrounding area, which forms a trough; and the area with the highest amount of lifting on the east side is the northern foothill of the small mountain range that extends to the east. Overall, the deformation at the base of the dam is primarily an A-LOS of about 40 mm, which is slightly greater than the A-LOS at the dam’s northern foot near the seepage interceptor and slightly less than that on the eastern slopes. In general, the deformation rate and deformation cumulative profile lines correspond to the profile lines, and the deformation characteristics of the profile line pathway area correspond to the deformation characteristics of the entire area in Figure 5.

Figure 11 illustrates the cumulative quantity of deformation versus time for the LOS deformation extreme and stabilization points in Figure 5. The highest value of M-LOS point is found at the midpoint between the top of dam and the junction of the dry beach. Based on analysis of satellite imagery, the maximum M-LOS value corresponds to the pipeline situated on the south side of the dam top (high-coherence point), which exhibits a uniform rate of M-LOS, with a cumulative value of 210 mm. We conclude that the material of the dam body and its own weight are the main causes of this area’s deformation. The dam body is primarily composed of the accumulation of coarse tailing sand, which has a large pore space, and the tailing sand has already completed the final stage of accumulation at the time of monitoring. Due to rain erosion, weathering, and other factors, the pore space and volume of the tailing sand decrease over time, causing the surface to subside. Not only that, but the hefty self-weight of the tailing sand caused the M-LOS to become bigger.

At the eastern end of the dam base, on top of the hill ridge, is where the maximum A-LOS point is found, with the summer of 2020 as the turning point, and the A-LOS speed is first fast and then slow, with a total A-LOS of about 60 mm. The elevation of the top of this hill ridge is about 164 m, that of the foot of the southern slope is 114 m, and that of the foot of the northern slope is 92 m, which forms a squeezing trend from south to north. We infer that the deformation in this region is primarily attributable to the mountain structure and accumulation of sediments. The small mountain range resembles an east–west wall, and the sediment accumulation on the south side is increasing, leading to the deformation of the mountain range on the north side and the deformation close to the LOS. Stabilization occurs at the junction of the dam’s top and base (shown as an orange dashed line in Figure 2), where deformation is more stable. Positively deformed regions are located far from the dam’s upper portion and are biased toward the dam’s base. We infer that the construction duration and materials of the tailings dam are primarily responsible for the stability of this region. The initial dam was built between 1988 and 1990 and was filled with moderately weathered kilolithic granite and covered with 110 m of clay. The stockpile dam was constructed from coarse sediment sand between 1991 and 2008. The junction between the top and bottom of the dam has an elevation of approximately 120 m, which is slightly higher than the summit of the stacked dam. We hypothesize that the formation of the stability zone is due to the early construction of the initial dam, the absence of seepage in the surface layer, and the relative stability of the dam body. In addition, the depth of the stockpile layer at the summit of the initial dam is less than ten meters, which is insufficient for the tailing sand to settle, so the surface is relatively stable and the stability zone is formed.

In the process of dam construction, coarse sand fed by the secondary cyclone gradually covered the initial dam and built the stockpile dam. However, due to the existence of the initial dam, the interior of the stockpile dam was divided into two regions, namely the upper portion of the dam body and the lower portion of the dam body, and the junction of these two regions was also the shortest distance between the top of the initial dam and the outer slope of the stockpile dam. The wastewater and tailings in the tailings pond exert continuous water and sediment pressure on the tailings dam towards the north, so the deformation at the bottom of the dam body is positive and close to the LOS direction, while the tailings dam has to bear its gravity, and the material of the outer slope is coarse sand, with large pores and no cover on the surface, which is affected by its gravity and the influence brought by the rainwater that continuously falls on the surface. The deformation is negative and distant from the direction of the line of sight.

Lyu (2019) compiled information on tailings dam failures around the globe and found that the primary causes of tailings dam failures are seepage, foundation failure, roofing, and earthquakes, with rainfall being the primary cause of the first three [79]. Duan (2023) analyzed the causes of tailings dam failures at a total of Shanxi Daoer Aluminum tailings dams using Sentinel-1 data. He discovered that the average deformational volume of the dams is related to the amount of rainfall, and that throughout the summer and fall, when rainfall is heavier and the tailings dam deforms more, the deformational volume of the dams rises. Changes in the water content of tailings ponds and tailings structures result in seasonal movements that have a substantial impact on slope stability [7]. Gama (2020) analyzed the deformation of the tailings ponds of the Córrego do Feijão mine in Brazil and the causes of the dam failure. He discovered that there is a relationship between changes in the amount of rainfall and the acceleration of the deformation displacements, with a smaller and stabilized rate of deformation during the dry period, and a larger and increasing rate of deformation during the abundant water period, and that the rainfall in the three days before the dam failure was the cause [30].

The climate of Dexing City is subtropical monsoon, with abundant precipitation, an average annual rainfall of 1800 mm, and moist springs and summers. The No.4 tailings pond has a total capacity of 835 million m^3^ and a catchment area of 14.3 km^2^, which is likely to cause the dam to fail if excessive rainfall occurs and cannot be eliminated in a timely manner. To investigate the relationship between tailings dam deformation and precipitation, we gathered the local monthly average rainfall during the deformation monitoring period and compared it to the sedimentation time series (Figure 11) [80,81]. The graph depicts an irregular distribution of precipitation throughout the year, with more precipitation in spring and summer and less in autumn and winter. Although rainwater runoff and infiltration will alter the expansion coefficient of the soil, which in turn affects the deformation characteristics of the dam body, deformation monitoring results indicate that there is no obvious correlation between the deformation amount and deformation rate of the No.4 tailings dam in the Dexing Copper Mine and the rainfall. We speculated that there were the following reasons:The drainage system of the tailings pond and tailings dam has played a role in timely evacuation of surface water, and the underground drainage and seepage pipes are also in normal operation, so there is no excess water storage in the tailings pond;The catchment area of the tailings reservoir is small, so the amount of water brought by rainfall is small, and it is difficult to have a significant impact on the tailings dam.

It can be assumed that the dam itself, and not rainfall, is accountable for the various deformation characteristics of the dam. However, the effect of precipitation on the deformation of the dams should not be disregarded, and the parameters influencing the operation of rainwater dams should be closely monitored.

### 5.3. Deformation Trend and Prediction

Depending on the time and material of dam construction, tailings dams can be divided into two portions (Figure 2): the internal initial dams and the later stockpile dams. Due to the fact that only stockpile dams can be observed at the surface, we only analyze the deformation trajectory of stockpile dams and predict their deformation over the next two years. According to our deformation monitoring data, the bottom of the dam is deforming in the opposite direction from the upper part of the dam. The upper part of the dam deforms away from the satellite direction, while the bottom of the dam deforms close to it, and the deformation trends are also quite different. Since the deformation of the dam body formed in two distinct directions, we selected all points within a radius of 50 m around the representative locations at the bottom and the top of the dam body, and averaged the deformation information of these points to obtain more representative deformation information. These data were utilized to analyze the time-series deformation trends of the dam body and to make reasonable predictions regarding future deformation.

General consolidation modeling is typically applied to flat areas, such as airports and land formations, that have undergone extensive and protracted artificial construction, including engineering operations such as infill and excavation, drainage, etc. These areas are relatively flat in the horizontal direction, and there are multiple layers in the vertical direction. The fill materials and geological conditions of each layer are quite distinct, and consolidation information can be obtained through geotechnical engineering monitoring. Hu et al. (2017) calculated the deformation of the tailings pond in Salt Lake County, Utah, using the multi-temporal InSAR method and data from multiple SAR sensors, and created a consolidation model of the tailings pond in Salt Lake County, Utah, using the specialized consolidation modeling and analysis software [34]. Xiong (2022) calculated the deformation of the tailings pond in Salt Lake County, Utah, based on the multi-temporal InSAR method to calculate the deformation of the reclaimed land area of Xiamen Xiang’an International Airport, using only Sentinel-1 data without the help of other SAR data and hydrological data. After comparing the fitted curves calculated by the hyperbolic model, Poisson model, and exponential model, it was found that the exponential curve model and the Asaoka method could better predict the final settlement and the consolidation time [34,82].

In general, these models are suitable for the regions with a flat surface and variations in deformation rate over the observation period, such as regions where the deformation velocity is first fast and then slow, with a tendency of 0 mm/yr. In contrast, the topography of our study area is more diverse, and the deformation speed remains single, and there is no sign of slowing down in the near future. Due to the quasi-linear deformation in these two places, the general consolidation model could not accurately fit the deformation time distribution characteristics. It is ultimately determined that the power function can maximally fit the deformation accumulation versus time; therefore, the deformation characteristics of the typical areas at the bottom and the top of the dam body were fitted and predicted based on the power function model (Figure 12). The R^2^ of the power function fitted curves was 0.9698 for the deformation of the lower portion of the dam and 0.9909 for the deformation of the upper portion of the dam. It is difficult to predict the end time of the deformation because the deformation rate of two parts of the dam body is relatively uniform during the observation time and there is no obvious variable speed. The power function fitting curve shows that the A-LOS rate of the bottom portion of the dam body is fast and then slow during the observation period, but it keeps a strong upward trend, and the annual A-LOS is about 10 mm. The M-LOS trend of the upper part of the dam body in the next two years is similar to the usual, with a quasi-linear trend M-LOS value of about 40 mm per year. In conclusion, although the change in the deformation rate of the two is not large, the change in the trend indicates that the deformation of the tailings dam may be developing in a more serious direction, and the tailings dam management personnel should formulate countermeasures.

## 6. Conclusions

In this study, we utilized the SBAS-InSAR method and 118 ascending Sentinel-1A images to investigate surface deformation around the No.4 tailings impoundment at the Dexing Copper Mine. Our findings reveal two distinct patterns of deformation in the tailings dam. The upper part of the dam experiences a significant move away from satellite LOS at a rate of approximately −40 mm/yr, while the bottom part exhibits a gradual approach to the satellite LOS, at a rate of 8 mm/yr. Notably, our analysis suggests that these deformation patterns are relatively independent of both temporal and spatial variations in precipitation. Instead, they may be influenced by factors such as the composition of tailings ponds and the construction materials used in the dam. The extensive size of the tailings pond, coupled with the substantial volume of tailings and wastewater it contains, results in distinct deformation patterns in the tailings dam. The lower section experiences extrusion and A-LOS, while the upper part undergoes significant M-LOS due to the coarse sand’s large pore size and the dam’s substantial self-weight. This research method enhances the repertoire of deformation monitoring techniques for the Dexing Copper Mine’s tailings dam and provides valuable long-term data for monitoring deformation. In summary, the safety of tailings dams significantly impacts both the mining operations and the well-being of nearby residents. Strengthening the monitoring of dam deformations is imperative to facilitate timely detection, early warnings, and prompt repairs. The deformation of Dexing Copper Mine’s No.4 tailings dam merits close attention from relevant authorities.

## Figures and Tables

**Figure 1 sensors-23-09707-f001:**
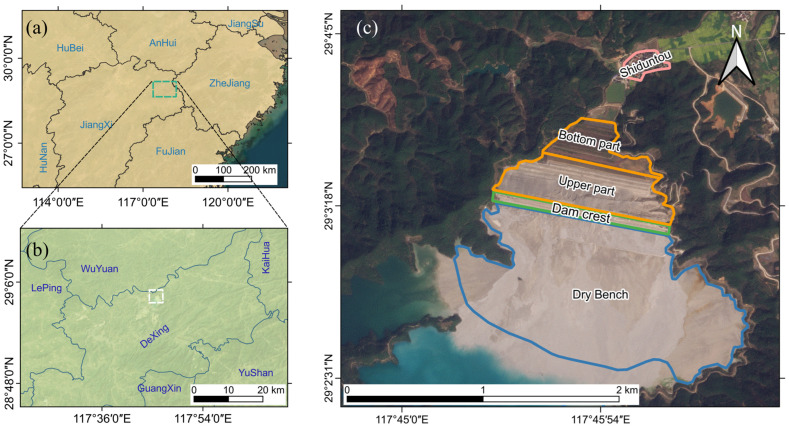
Study area. (**a**) The location of Dexing on the map. (**b**) The location of the No.4 tailings pond on the map. (**c**) The optical satellite map of the study area. The black line of (**a**) represents the provincial boundary. The range shown in the green rectangle of (**a**) is consistent with that in (**b**). The white rectangle in (**b**) indicates the extent of (**c**). Satellite image from Planet [65].

**Figure 2 sensors-23-09707-f002:**
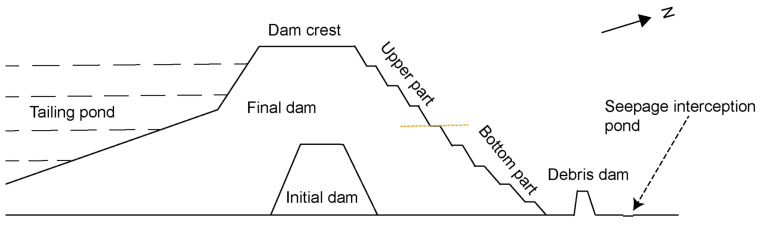
Schematic diagram of the dam profile of the No.4 tailings pond. The profile from left to right corresponds to the orientation from south-southeast to north-northeast. The orange dashed line indicates the dividing line between the upper and bottom parts.

**Figure 3 sensors-23-09707-f003:**
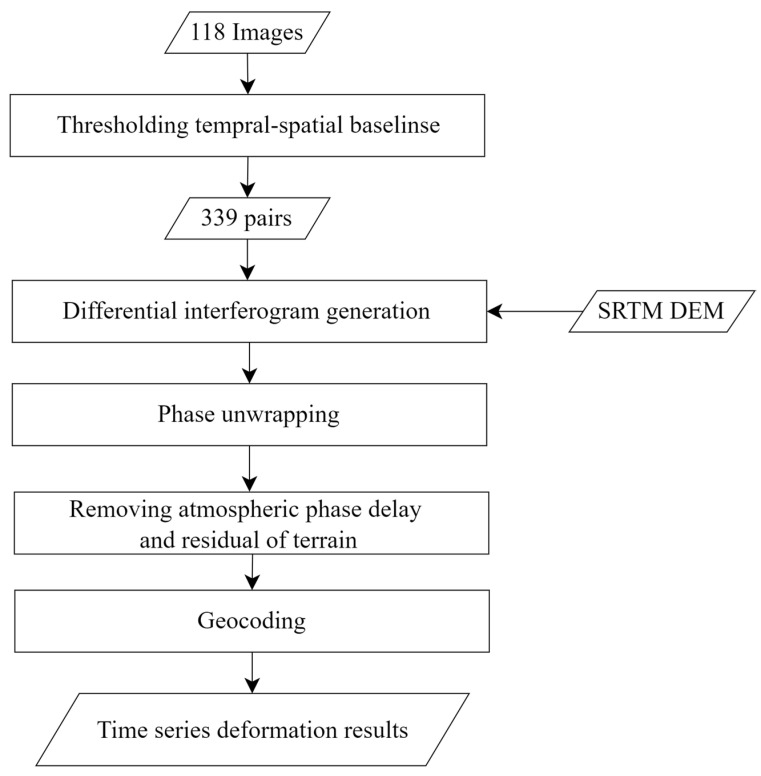
Flowchart of the SBAS-InSAR data processing.

**Figure 4 sensors-23-09707-f004:**
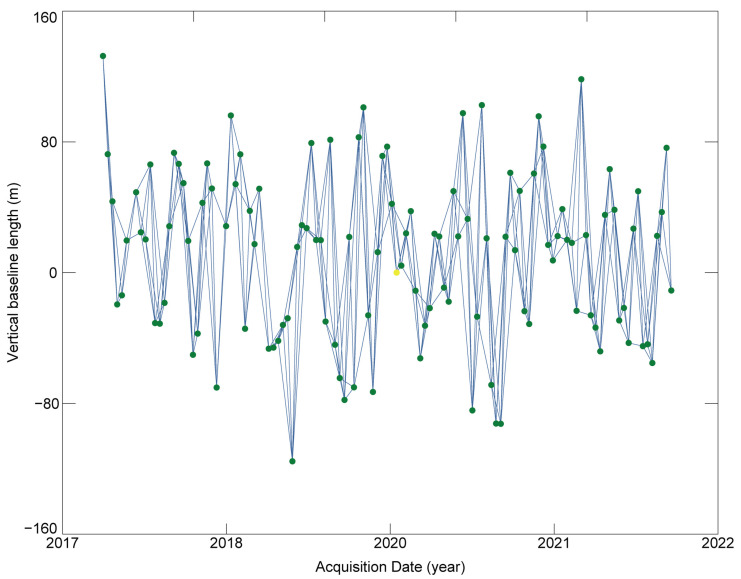
Spatio-temporal baseline network. The green dot shows the obtained image, and each blue line between two distinct locations represents an interferogram pair. The yellow dot corresponds to the co-registration reference image.

**Figure 5 sensors-23-09707-f005:**
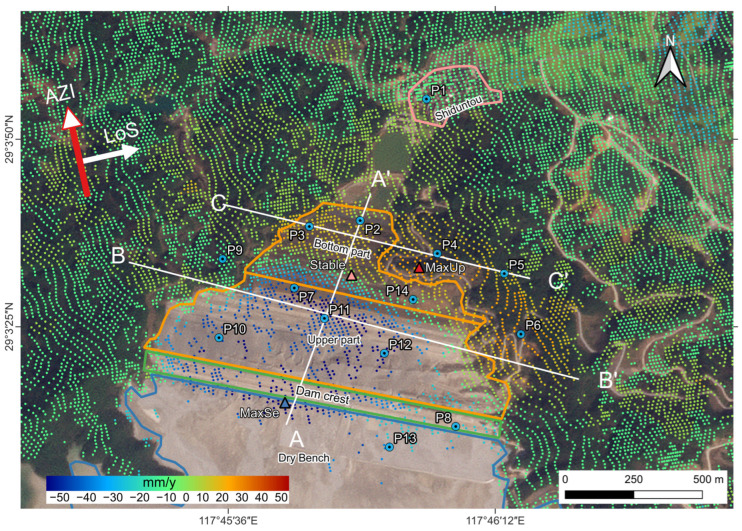
The No.4 tailings pond dam and adjacent deformation rate map. The dry beach, the top of the dam, and the outer slope, the seepage interceptor pool, and the village of Shiduntou are shown from south to north in the figure. The red, blue, and yellow triangles represent the positive and negative extremes of the LOS deformation of the dam body, as well as the relative stabilization point, respectively. P1–P14 are selected specific deformation points; A, A’, B, B’, C, C’ are the starting point of three cross-sectional profiles.

**Figure 6 sensors-23-09707-f006:**
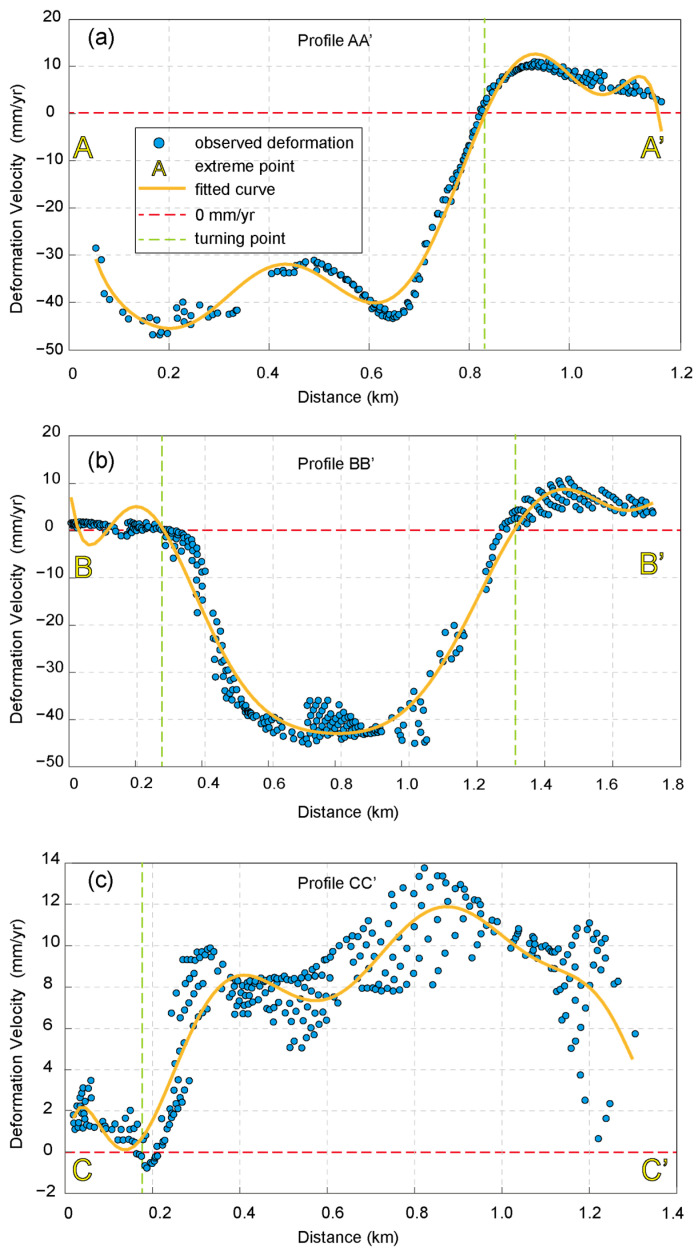
Deformation rate profiles. Deformation rate profiles for (**a**) line AA’, (**b**) BB’, and (**c**) CC’. The blue dot represents the average deformation rate at all sites within 30 m of the buffer zone of the line segment. The orange line represents the deformation rate fitting curve, the red dashed line represents a deformation rate of 0 mm/yr, and the green dashed line represents the location where the deformation rate is 0 mm/yr.

**Figure 7 sensors-23-09707-f007:**
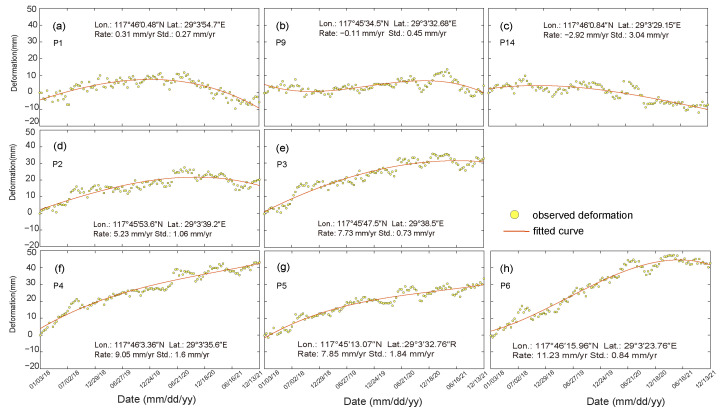
Cumulative deformation of P1–P6 (**a**,**d**,**e**–**h**), P9 (**b**), and P14 (**c**), their distribution positions are shown in Figure 5. The yellow point indicates the average deformation rate at all points within a 30 m radius centered on the point.

**Figure 8 sensors-23-09707-f008:**
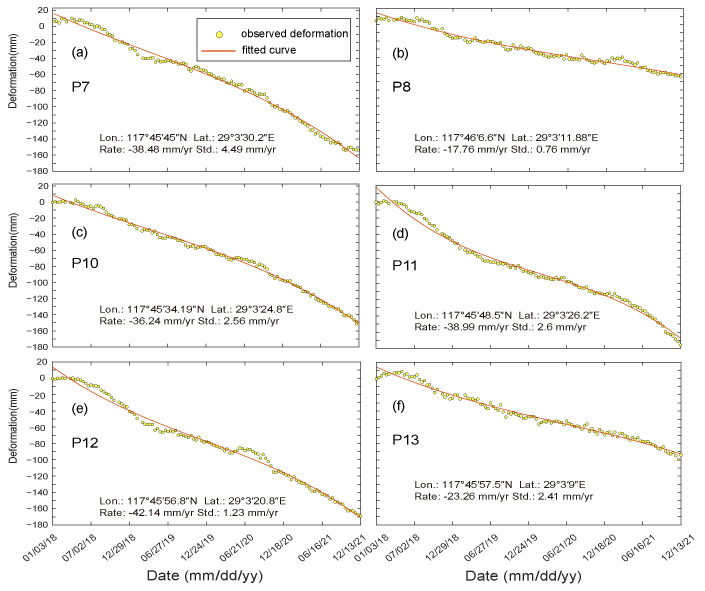
Cumulative deformation of P7 (**a**), P8 (**b**), and P10–P13 (**c**–**f**), their distribution positions are shown in Figure 5. The yellow point indicates the average deformation rate at all points within a 30 m radius centered on the point.

**Figure 9 sensors-23-09707-f009:**
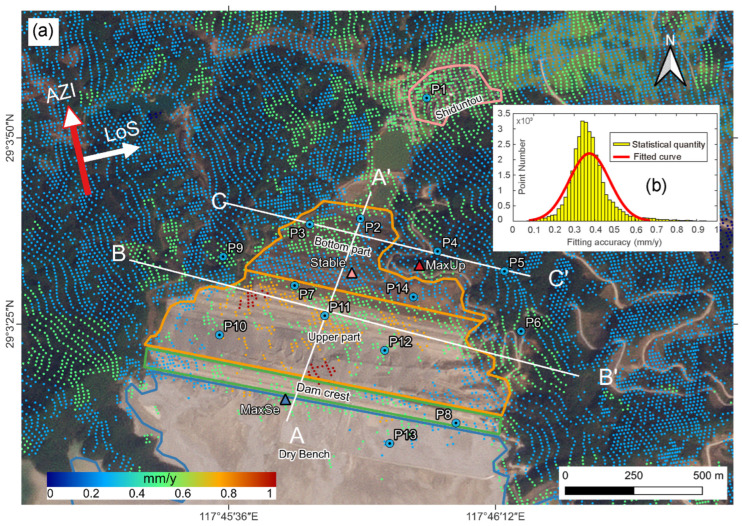
Precision and statistical chart of LOS deformation. (**a**) The figure of fitting accuracy. (**b**) The distribution of fitting accuracy. P1–P14 are selected specific deformation points; A, A’, B, B’, C, C’ are the starting point of three cross-sectional profiles.

**Figure 10 sensors-23-09707-f010:**
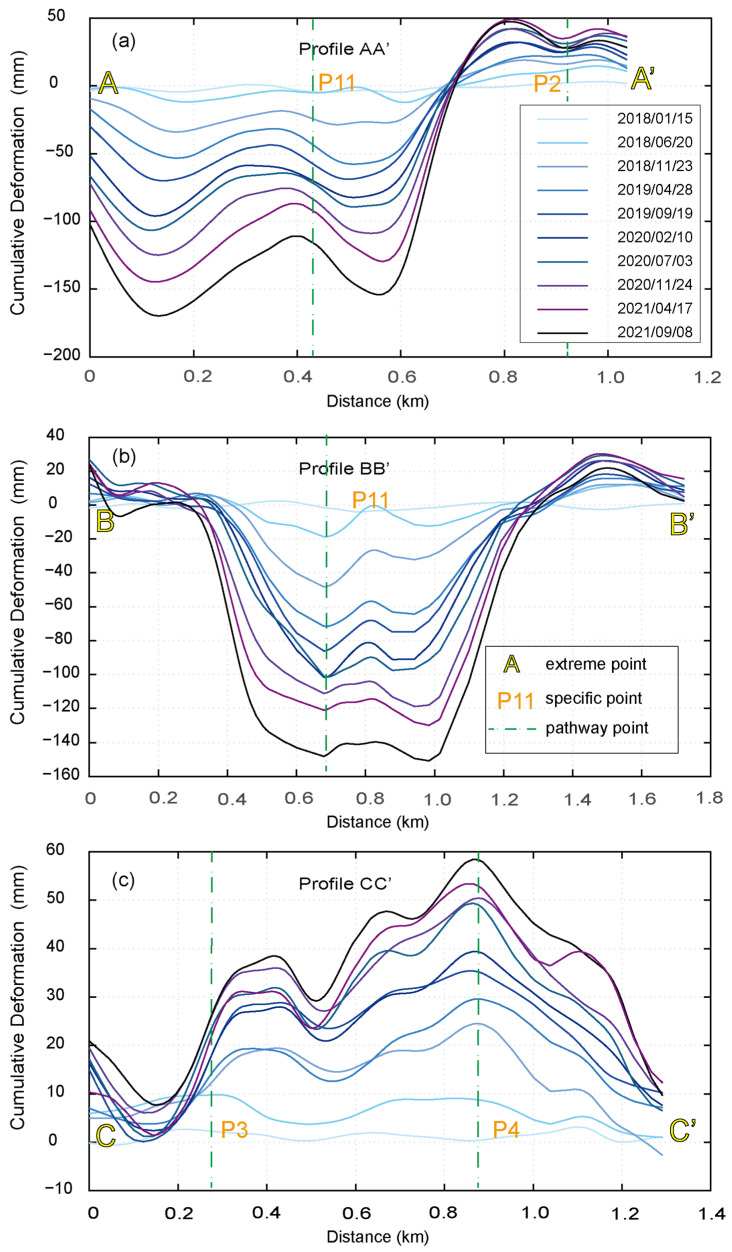
Fitting diagrams of deformation accumulation. (**a**) Deformation accumulation fitting plots for the AA’ line pathway region. (**b**) Deformation accumulation fitting plots for the BB’ line pathway region. (**c**) Deformation accumulation fitting plots for the CC’ line pathway region. Sub-temporal deformation accumulation for the regions of the line AA’, BB’, and CC’ pathways, where the green dashed line travels through the deformation accumulation at the deformation observation point depicted in orange text.

**Figure 11 sensors-23-09707-f011:**
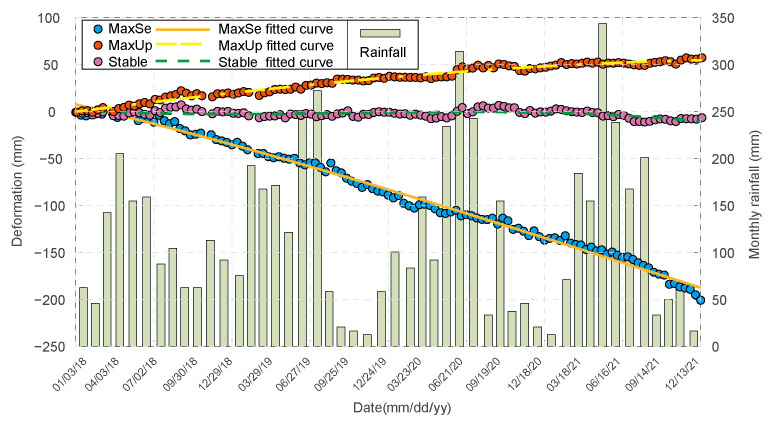
A scatterplot of dam deformation accumulation versus precipitation. The red, blue, and pink points depict the deformation accumulation at the point of maximum A-LOS, maximum M-LOS, and stabilization, respectively, while the aqua-green bar represents the current month’s precipitation. The duration is between January 2018 and December 2021.

**Figure 12 sensors-23-09707-f012:**
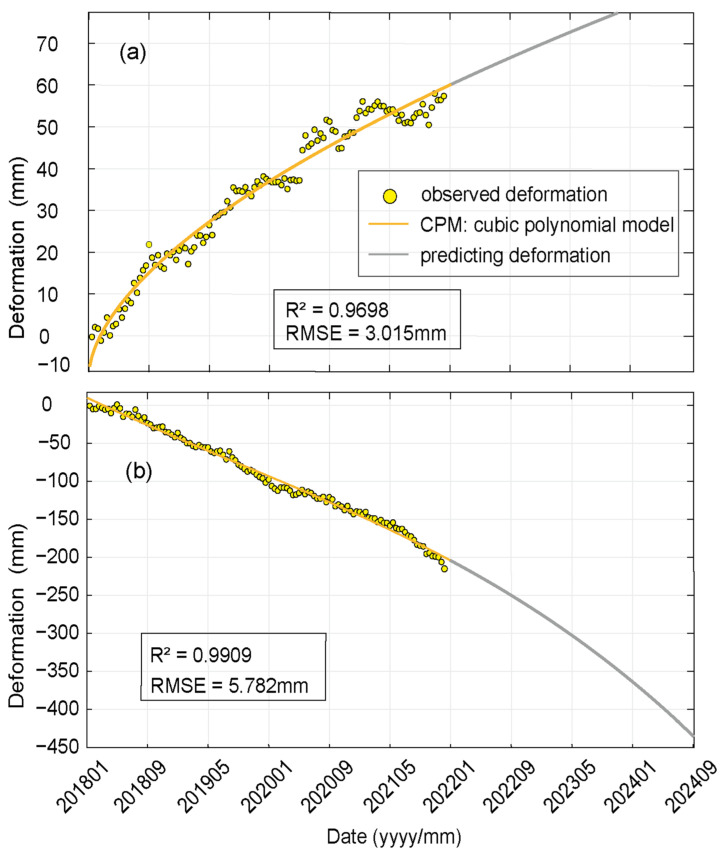
Power function model fitting curves for the bottom and upper deformation of the dam body. (**a**) power function fitting curves of deformation at the bottom of the dam body, (**b**) power function fitting curves of deformation at the upper part of the dam body. The yellow line is the three fitting curves, and the gray line represents the predicted deformation.

**Table 1 sensors-23-09707-t001:** Summary of tailings dam failure incidents from in recent years.

Date	Location	Ore Type	Probable Cause of Failure
31 January 2023	Kearl oil sands mine, Fort Chipewyan, Alberta, Canada	bitumen	process water drainage pond overflow [11]
7 November 2022	Williamson Mine, Mwadui Lohumbo, Kishapu District, Shinyanga Province, Tanzania	diamond	tailings dam failure, 150 m wide breach of the eastern wall of the impoundment [12]
11 September 2022	Jagersfontein, Kopanong, Xhariep, Free State, South Africa	diamond	high water level in the tailings reservoir destroyed the stability of the dam body [12]
23 July 2022	Agua Dulce, Potosí, Bolivia	silver, zinc	illegal mineshaft [13]
27 March 2022	Wenquan Township, Jiaokou County, Shanxi Province, China	bauxite	the rainfall reduced the stability of the dam and caused tailings dam failure [7]
20 January 2022	Banjhiberana village, Thelkoloi area, Sambalpur district, Odisha (formerly Orissa), India	iron	breach of tailings pond wall holding iron slurry generated from beneficiation plant [14]
8 January 2022	Pau Branco mine, Nova Lima, Minas Gerais, Brazil	iron	heavy rain leads to the collapse of the tailings pond slope and the overtopping of the tailings dam [15]
26 November 2021	San Antonio de María mine, Ananea, San Antonio de Putina province, Puno, Peru	gold	tailings dam (settling pond) failure after heavy rain [16]
27 July 2021	Catoca mine, Saurimo, Lunda Sul, Angola	diamond	breach in spillway duct leads to massive spill of “rejected pulp” [17,18]
2 July 2020	Hpakant, Kachin state, Myanmar	jade	waste heap failure [19]
28 March 2020	Tieli, Yichun City, Heilongjiang Province, China	molybdenum	“No.4 overflow well” of the tailings dam tilted, resulting in the release of supernatant water and tailings through a drainage tunnel [20]
10 July 2019	Cobriza mine, San Pedro de Coris district, Churcampa province, Huancavelica region, Peru	copper	tailings dam failure [21]
9 April 2019	Muri, Jharkhand, India	bauxite	failure of red mud tailings pond [22]
25 January 2019	Córrego de Feijão mine, Brumadinho, Região Metropolitana de Belo Horizonte, Minas Gerais, Brazil	iron	seepage erosion piping, weakening of the structure [23]
10 July 2019	Cobriza mine, San Pedro de Coris district, Churcampa province, Huancavelica region, Peru	copper	tailings dam failure [24]

**Table 2 sensors-23-09707-t002:** Dam Information.

Dam	Height	Top Width	Construction Material	Construction Method	Outside Slope Ratio	Construction Time
Initial dam	38 m	10 m	Chimneystone	centerline embankment method	1:3	1988–1990
Final dams	208 m	40 m	Coarse Tailings		1:3.5	1991–2008

**Table 3 sensors-23-09707-t003:** The main parameters of SAR image.

Parameter	Value
Beam mode	Sentinel-1A IW
File type	L1 Single Look Complex (SLC)
Orbit number	Path-142, Frame-91
Central incidence angle (°)	39.369
Orbit direction	Ascending
Ground swath WIDTH	250 km
Resolution (range and azimuth)	5 × 20
Polarization	VV, HV
Temp. Res (Day)	12
Satellite transit time (UTC)	10:10

## Data Availability

The Sentinel-1 images are from the Copernicus Open Access Hub (https://scihub.copernicus.eu/, accessed on 3 June 2023) of the European Space Agency (ESA). The Monthly average rainfall data from A Big Earth Data Platform for Three Poles (TPDC, http://poles.tpdc.ac.cn/, accessed on 5 June 2023). The base maps are from Planet https://www.planet.com/explorer/ (accessed on 11 November 2023).

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
