# Peer review of "SBAS-InSAR Based Deformation Monitoring of Tailings Dam: The Case Study of the Dexing Copper Mine No.4 Tailings Dam"

_sensors, 2023, doi:10.3390/s23249707_

Round 1

Reviewer 1 Report

Comments and Suggestions for Authors

This is a very practical and much needed topic area. The authors should be congratulated for undertaking this research.

Unfortunately, the manuscript doesn't introduce significant new knowledge needed to address the problem of tailings dam monitoring. Many companies already provide this type of service, although the findings are seldom published due to confidentiality clauses. The manuscript submitted by the authors can form a basis for a more elaborate and practical document.

Regarding the analytical technique used by the authors, I have the following comments:

- You indicate that you could only acquire the ascending mode data, yet you provide definitive conclusions regarding up/down surface displacements. Are you certain that descending data is not really needed to substantiate your conclusions and resolve the displacement components? Sometimes these composite movements can be deceiving and relying just on one look direction can create false impression. You indicate that the deformation in the study area is mainly vertical (page 9, line 266). Why are you so sure about it?

- You report displacement rates based on your InSAR analysis. Is there any independent ground truth measurement that verifies at least some of your conclusions?

- You state in the abstract that "the research findings can serve as a decision-making basis" (I would say tool), but your conclusions do not provide any decision-making tree. We really need some criteria to make these important safety-related decisions. For example, what rate of movement would trigger an alarm level?

I think the manuscript can be greatly improved by expanding the geotechnical and geology information about the site. It needs to be written in a language commonly used by geotechnical engineers. I suggest expanding on site characterization, dam construction (material and methods used in construction) and relevant heights. In other words, we need more detail about the structure that we are monitoring.

- One of the main shortcomings is the tailing material characterization. Assuming that the behavior of copper tailings differs from other types, I suggest providing additional data, such as:

- Specific gravity

- Unit weight

- pH

- Angle of repose

Also, it would be extremely informative to compile published reports of recent tailings dam failures in a concise tabular form, listing suspected causes of failure. We often learn from failures.

I suggest expanding the manuscript along the suggestions listed above and then resubmitting. It has the potential of becoming a valuable resource to the practitioners.

Comments on the Quality of English Language

English language related to the site geology and geotechnical matters should be improved.

Reviewer 2 Report

Comments and Suggestions for Authors

Manuscript ID sensors-2697375 entitled "Research on deformation of Dexing tailings dam based on Time-Series InSAR technology" has been reviewed. In this study, the authors employed synthetic aperture radar (SAR) interferometry, with a specific emphasis on the small baseline subset (SBAS) InSAR technique, to delve into the deformation characteristics of the No. 4 tailing dam situated in the Dexing Copper Mine, Jiangxi Province in China.The study utilized Sentinel-1 datasets to extract line-of-sight (LOS) surface deformation. The findings presented in this study carry significant implications for enhancing the safety monitoring of tailings dams. The surface deformation data obtained by SAR satellite has the characteristics of high monitoring efficiency and wide monitoring space and time range, which provides a research idea different from the traditional perspective and method, and provides a good idea for the deformation monitoring of the same type of infrastructure. The manuscript is clearly articulated, well-written with clear expression. I recommend accepting this article following the  minor revision recommendations :

# Major comments:

1. The study area is located in the subtropical mountains, and there is a significant presence of tropospheric delay and atmospheric water vapor in the data over a three-year timeframe. How was the phase interference from these elements mitigated?

2. Rainfall data shows obvious seasonal patterns, but the paper doesn't establish a correlation between rainfall and dam deformation. Could you clarify the rationale behind comparing rainfall data with deformation monitoring data?

#Specific comments:

Lines 14-15, “The tailing pond is a man-made debris flow hazard with high potential energy”, have an ambiguity, the words "source" or "area" should be added.

Line 91, there's an extra period.

Lines 85-87, the cohesion does not flow smoothly, a conjunction should be added before Surface deformation.

Lines 88-91, Sentences that are too long need to be condensed or broken into shorter sentences.

Lines 111-113, few should be changed to a few.

Lines 138-148, the content is slightly similar to that in Figure 1, and there is no explanation for the area of pink lines in the figure, so “shiduntou” can be changed to village.

Lines 156-178, the description of the dam body is too long and complicated, and it needs to be simplified and focused.

Lines 187-197, the description of the parameters of the Sentinel satellite is provided in the table, which can be appropriately condensed in the manuscript.

Line 205, word error: bilinear interpolation rather than difference.

Lines 211-212, the wording was Thoroughly repetitive.

Lines 222&231, the interpretation of LOS is different, please check.

Line 273, there's an extra period.

Line 341-342, In Figure 7, this point refers to the point in the upper left corner of the small diagram and should be written clearly.

Line 357, After you have explained the formula, add a new paragraph under "according".

Lines 561-564, check that "bottom" is written twice in the description field in Figure 12.

Comments on the Quality of English Language

Manuscript ID sensors-2697375 entitled "Research on deformation of Dexing tailings dam based on Time-Series InSAR technology" has been reviewed. In this study, the authors employed synthetic aperture radar (SAR) interferometry, with a specific emphasis on the small baseline subset (SBAS) InSAR technique, to delve into the deformation characteristics of the No. 4 tailing dam situated in the Dexing Copper Mine, Jiangxi Province in China.The study utilized Sentinel-1 datasets to extract line-of-sight (LOS) surface deformation. The findings presented in this study carry significant implications for enhancing the safety monitoring of tailings dams. The surface deformation data obtained by SAR satellite has the characteristics of high monitoring efficiency and wide monitoring space and time range, which provides a research idea different from the traditional perspective and method, and provides a good idea for the deformation monitoring of the same type of infrastructure. The manuscript is clearly articulated, well-written with clear expression. I recommend accepting this article following the  minor revision recommendations :

# Major comments:

1. The study area is located in the subtropical mountains, and there is a significant presence of tropospheric delay and atmospheric water vapor in the data over a three-year timeframe. How was the phase interference from these elements mitigated?

2. Rainfall data shows obvious seasonal patterns, but the paper doesn't establish a correlation between rainfall and dam deformation. Could you clarify the rationale behind comparing rainfall data with deformation monitoring data?

#Specific comments:

Lines 14-15, “The tailing pond is a man-made debris flow hazard with high potential energy”, have an ambiguity, the words "source" or "area" should be added.

Line 91, there's an extra period.

Lines 85-87, the cohesion does not flow smoothly, a conjunction should be added before Surface deformation.

Lines 88-91, Sentences that are too long need to be condensed or broken into shorter sentences.

Lines 111-113, few should be changed to a few.

Lines 138-148, the content is slightly similar to that in Figure 1, and there is no explanation for the area of pink lines in the figure, so “shiduntou” can be changed to village.

Lines 156-178, the description of the dam body is too long and complicated, and it needs to be simplified and focused.

Lines 187-197, the description of the parameters of the Sentinel satellite is provided in the table, which can be appropriately condensed in the manuscript.

Line 205, word error: bilinear interpolation rather than difference.

Lines 211-212, the wording was Thoroughly repetitive.

Lines 222&231, the interpretation of LOS is different, please check.

Line 273, there's an extra period.

Line 341-342, In Figure 7, this point refers to the point in the upper left corner of the small diagram and should be written clearly.

Line 357, After you have explained the formula, add a new paragraph under "according".

Lines 561-564, check that "bottom" is written twice in the description field in Figure 12.

Reviewer 3 Report

Comments and Suggestions for Authors

The paper highlights the efficiency of SBAS in monitoring deformation in the tailings dam. Surface deformation in the area of interest is thoroughly investigated. My comments and suggestions are as follows:

Line 48: please remove the second the

Line 87: monitired->monitored

Line 89: by temporal

Line 90: please remove the second dot after deformation extraction

Line 99: you could cite a few papers for SBAS applications

e.g.

Sun, H.; Peng, H.; Zeng, M.; Wang, S.; Pan, Y.; Pi, P.; Xue, Z.; Zhao, X.; Zhang, A.; Liu, F. Land Subsidence in a Coastal City Based on SBAS-InSAR Monitoring: A Case Study of Zhuhai, China. Remote Sens. 2023, 15, 2424. https://doi.org/10.3390/rs15092424

Kulsoom, I., Hua, W., Hussain, S. et al. SBAS-InSAR based validated landslide susceptibility mapping along the Karakoram Highway: a case study of Gilgit-Baltistan, Pakistan. Sci Rep 13, 3344 (2023). https://doi.org/10.1038/s41598-023-30009-z

Xiao, B.; Zhao, J.; Li, D.; Zhao, Z.; Zhou, D.; Xi, W.; Li, Y. Combined SBAS-InSAR and PSO-RF Algorithm for Evaluating the Susceptibility Prediction of Landslide in Complex Mountainous Area: A Case Study of Ludian County, China. Sensors 2022, 22, 8041. https://doi.org/10.3390/s22208041

Pang, Z.; Jin, Q.; Fan, P.; Jiang, W.; Lv, J.; Zhang, P.; Cui, X.; Zhao, C.; Zhang, Z. Deformation Monitoring and Analysis of Reservoir Dams Based on SBAS-InSAR Technology—Banqiao Reservoir. Remote Sens. 2023, 15, 3062. https://doi.org/10.3390/rs15123062

Line 109: please remove the second and

Line 203: The rainfall data covers the same time period as InSAR data?

Line 215: Do you mean phase unwraping?

Line 234: mainly instead of manily

Line 272: please remove the second dot

Round 2

Reviewer 1 Report

Comments and Suggestions for Authors

This version is a significant improvement over the original. It still requires extensive English editing for improved readability prior to publishing. The abstract in particular doesn't read well. Also, I suggest that the authors consider alternate title in order to capture the essence of their contribution. One possible suggestion - Monitoring deformations of copper tailings dam using satellite remote sensing.

With regard to the InSAR analysis, I still don't understand how the authors can assert vertical settlement and uplift based just on the ascending data processing. Perhaps the word "apparent" would qualify that assessment. You are interpreting LOS results. Typically when we look at the ground movement around a hillside, it tends to be compound rather than just vertically up/down.

Table 1 provides an important source of information but can you elaborate? Your last column is titled 'Phenomena and causes of accidents' (I suggest you simply call it 'Probable cause of failure'. The problem is that you have many instances of 'tailing dam failure' without listing the cause. This is not very helpful. Try to update the missing info. One thing that strikes me is the frequent reference to 'heavy rain'. Could this be the most common denominator? If so, how do you handle it at your site?

Your subsequent discussion suggests than rainfall is not a major factor. Perhaps not in this particular case. Please elaborate on the drainage system used on this site. Perhaps a separate figure showing its layout would be beneficial. I think it may contain ideas that can be adopted in other projects (assuming it functions as intended). Also, show the details of the seepage interception pond.

The text starting on line 107 (The No. 4...) up to line 116 should be the first paragraph of the Study Area section (starting on line 124).

You indicate highly alkaline tailing and highly acidic mine wastewater. Please elaborate on how these factors were accounted for in the dam design. For example, was the dam/soil permeability assessed using the actual wastewater?  Please elaborate on these design challenges.

Your Table 2 shows dam information. You refer to the 'initial dam' and 'accretion dams'. I suggest you label these particular features on Figure 2. Also, show some critical dimensions on Figure 2. Is the height really 208 meters?

Text on line 505 and 506 - "...expansion coefficient of the soil".  I don't understand what you mean.

Text on line 507 and 508 - "...deformation monitoring results indicate that there is no correlation between the deformation volume (?) and rate of deformation of the dam body and the amount of rainfall".  If you really feel confident about this statement then it should be listed as one of your conclusions but probably narrowly qualified by saying "for this particular site". Don't make it an overly general conclusion.

I think the plots shows on Figure 11 and 12 are valuable but have you considered presenting your data in other forms? Perhaps using a logarithmic scale, somewhat similar to what we do for soil consolidation testing. Also, there have been attempts to present rock slope InSAR data in a manner that allows extrapolation to the time of failure. I'm just suggesting that you explore ways of presenting your data in a manner that indicates future trend or change of trend. Anything that would help in decision making would be helpful. Ultimately this manuscript should project your expertise in analyzing a tailings dam behavior and offer an idea of how to implement an early warning system.

Comments on the Quality of English Language

Extensive English editing will be required at the final stage.
